# Mid-Air Gestural Interaction with a Large Fogscreen

**Vera Remizova** [1,*] **, Antti Sand** [1] **, I. Scott MacKenzie** [2] **, Oleg Špakov** [1] **, Katariina Nyyssönen** [3] **,
Ismo Rakkolainen** [1] **, Anneli Kylliäinen** [3] **, Veikko Surakka** [1] **and Yulia Gizatdinova** [1]

[1] Faculty of Information Technology and Communication Sciences, Tampere University,
33100 Tampere, Finland
[2] Department of Electrical Engineering and Computer Science, York University, Toronto, ON M3J 1P3, Canada
[3] Psychology Clinic PSYKE, Faculty of Social Sciences, Tampere University, 33100 Tampere, Finland
* Correspondence: vera.remizova@tuni.fi

**Abstract:** Projected walk-through fogscreens have been created, but there is little research on the evaluation of the interaction performance with fogscreens. The present study investigated mid-air hand gestures for interaction with a large fogscreen. Participants ($N = 20$) selected objects from a fogscreen using tapping and dwell-based gestural techniques, with and without vibrotactile/haptic feedback. In terms of Fitts' law, the throughput was about 1.4 bps to 2.6 bps, suggesting that gestural interaction with a large fogscreen is a suitable and effective input method. Our results also suggest that tapping without haptic feedback has good performance and potential for interaction with a fogscreen, and that tactile feedback is not necessary for effective mid-air interaction. These findings have implications for the design of gestural interfaces suitable for interaction with fogscreens.

**Keywords:** touchless user interface; mid-air hand gestures; fogscreen; haptic feedback; Fitts' law





## 1. Introduction

Immaterial mid-air displays formed from flowing light-scattering particles (usual fog) bring new possibilities for displaying information. An interactive fog display—hereafter "fogscreen"—is a walk-through, semi-transparent display onto which interactive objects are projected (see Figure 1). Fogscreens, being transparent and penetrable yet physical interaction mediums, are most engaging when users can directly interact and play with them, gaining a range of futuristic experiences, such as mid-air gaming, cutting-edge digital signage, and engaging walk-through stereoscopic visualizations. Walk-through fogscreens are commonly employed in various settings, such as trade shows, theme parks, museums, and concerts. Equipped with multiple sensors, fogscreens can act as an interactive touchscreen or walk-through virtual reality screen [1–3].

The fogscreen used in this study consists of a non-turbulent fog flow inside a wider non-turbulent airflow [4]. Technical details of our fogscreen are given in Section 3. In contrast to most physical user interfaces, however, fogscreens have no pronounced tactile feedback [5]. A fogscreen user may feel a slight sensation of the air, some humidity, and coldness of the fog, but no tactile feedback that would resemble a touchscreen.

Despite the usage potential of fogscreen technology and a great number of prototypes having recently been presented, to our surprise, no user studies have been conducted with fogscreens to collect systematic evidence about their interaction properties. The lack of understanding of design requirements for mid-air interaction with fogscreen devices hinders the development of user interfaces specifically tailored for this technology. As our research shows, interaction with fogscreens extends from full-body motion (such as walking or jumping through the fog) to fine hand movements (such as physically clicking on the fog). The latter appears especially interesting, as it extends the functionality of fogscreens towards being comparable with conventional touchscreens in terms of object manipulation, while preserving the benefits of touchless mid-air interaction.

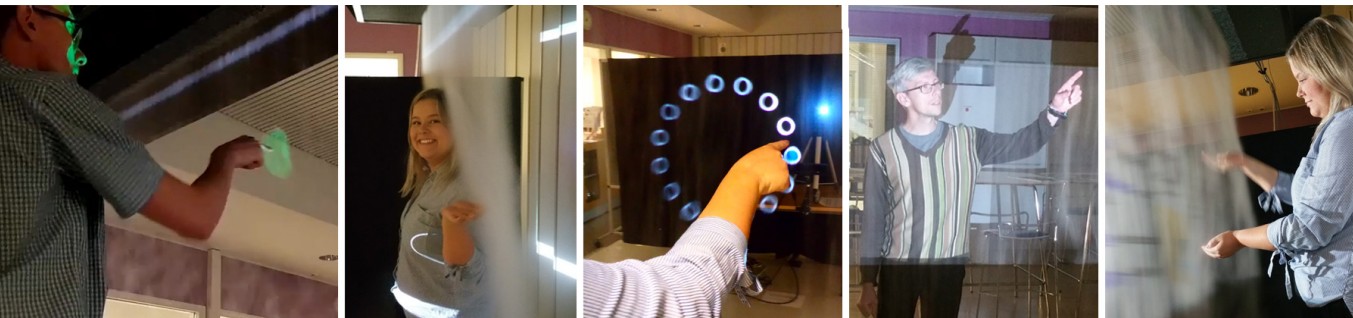

**Figure 1.** Participants interacting with a fogscreen.

In this study, two mid-air gestures for physical interaction with a large fogscreen were examined, namely, *tapping* and *dwell-based* selection gestures. Both were implemented in a way that involved physical interaction with the fog. For tapping in the fog, the hand movement imitates a conventional physical tap on a vertical screen. For dwell-based selection in the fog, a finger dwells in the fog for one second. The difference between the gestures is that the dwell-based selection gesture does not require removing the user's hand from the fog. The user may continuously point and select targets while their hand travels from one target to another in the fog. Conversely, fog tapping requires the hand to leave the fog to register a target selection.

Visual feedback alone may not be effective for interacting with large fogscreens due to slight turbulences in the fog that can blur projected graphics and poor lighting conditions that make it difficult for users to see whether the interaction is performed correctly [6]. To address this issue, we investigated the effects of haptic feedback on user performance and subjective experiences during object selections with a large fogscreen. The two gestures (dwell-based and tapping) were implemented with different feedback modalities (auditory, visual, and haptic). A custom-built lightweight wireless vibrotactile actuating device located on the user's hand was used for haptic feedback.

Thus, the aim of this study was to systematically investigate whether there were empirical differences between the two selection methods with different vibrotactile feedback in a number of target acquisition tasks with a large fogscreen. More specifically, the contribution of this work includes (a) a comparison of Fitts' law performance of tapping and dwell-based gestures in a number of target acquisition tasks, (b) quantifying the effects of the gestures and haptic feedback on user performance and subjective experience, and (c) discussing the applicability of the gestures for interaction with a large fogscreen and emphasizing factors for designing such systems.

## 2. Related Work

### 2.1. Particle Displays

Images have been projected on various surfaces, such as water, smoke, or fog, for more than 100 years (e.g., the patent by Just [7]). Modern fogscreens are concise, thin-particle projection systems producing an image quality superior to previous systems [8]. The screen is dry, cool to the touch, and permeable, so it is possible to physically reach or walk through the image. At the same time, since the fog flow is not perfectly flat, small movements of the fog due to turbulence occur, which in turn cause a slight movement in the projected graphics. Fogscreens have several benefits, including their ability to automatically and instantly recover when penetrated, as well as their ability to remain clean and hygienic as there is no surface for dirt, bacteria, or viruses to transfer. This opens new possibilities for their use [9]. Fogscreens can also be dormant and spring into action as users approach them [2].

Early studies and experiments indicate that fogscreens have immense potential as an interaction medium, offering unique and innovative ways for people to interact with digital content and technology [3,10]. Fogscreens are most engaging when the audience can directly interact and play with them. For instance, physical interaction with the fog

increases children's motivation and engagement in gaming [1]. Additionally, physical transparent screens enhance collaboration by enabling face-to-face co-location of multiple users [11].

### 2.2. Mid-Air Interaction with Large Conventional Screens

The emergence of depth-sensing methods that track the 2D/3D position of the hand has enabled the development of mid-air hand interaction for large screens, and while interaction with fog is still largely an under-investigated area, significant progress was achieved in mid-air hand gesturing with large conventional projector displays. Table 1 summarizes the findings on the performance of mid-air interaction techniques for target selection. Different researchers favored different gestures for selection tasks, making a choice based on the gestures' salient characteristics. For example, dwell-based interaction is intuitive and accurate [12–14], whereas tapping gestures are fast and consistent with touchscreen interaction [15–17]. Some researchers combined mid-air gestural techniques with other modalities, such as the voice [18] or physical buttons [19]. Furthermore, a user preference for tactile feedback during interactions with touchless user interfaces via gestural inputs has been reported by other studies [15,20].

**Table 1.** Summary of the performance evaluation for mid-air interaction gestures in target selection.

| Reference | Interaction Gestures * | Throughput ** (bps) | Results | Technology | Application Domain |
|---|---|---|---|---|---|
| Dube et al. [15] | Dwell-based (800 ms) | 1.73 | Dwell-based gesture was slowest, but the most accurate and the least physically and cognitive demanding. | Leap Motion Controller | Desktop |
| | Dwell-based (800 ms) + haptic | 1.74 | | | |
| | Tapping | 1.75 | | | |
| | Tapping + haptic | 2.07 | | | |
| Pino et al. [18] | Gesture + voice | 2.1 | Gesture interaction was slower and harder compared to using a mouse. | Microsoft Kinect Sensor | Desktop |
| Schwaller and Lalanne [21] | Dwell-based (500 ms) | 1.9 | The performance of dwell-based gesture was lower than that of pinching, but dwell-based was more accurate. | Microsoft Kinect Sensor | Large screen |
| Hespanhol et al. [13] | Dwell-based | – | Dwell-based was the most intuitive gesture for selection; tapping was a common gesture in other digital interfaces. | Microsoft Kinect Sensor | Large screen |
| | Tapping | – | | | |
| Burno et al. [19] | Gesture S + button | 1.9 | Gesture interaction had lower throughput compared to using a mouse and a touchscreen. | Leap Motion Controller | Desktop |
| | Gesture M + button | 1.9 | | Creative Senz3D Camera | Desktop |
| | Gesture L + button | 2.25 | | PrimeSense Carmine 1.09 3D Camera | Desktop |
| Erazo et al. [12] | Tapping | – | Dwell-based gesture was the most intuitive and accurate for selection. | Microsoft Kinect Sensor | Large screen |
| | Dwell-based (500 ms) | – | | | |

* The table uses unified names for mid-air gestural techniques. *Tapping* refers to a pushing forward gesture moving a certain distance, which is also known as 'push' or 'tap' in the literature. *Dwell-based* signifies holding the hand on a selectable element for a fixed time, which is also referred to as 'dwell' or 'hover.' *Gesture +* denotes hand motions for target selection and other input modalities for confirmation. ** Note that the software and hardware used, as well as the specific Fitts' law variant employed, can vary across different studies and may affect the calculated throughput values.

Some technological challenges were identified for mid-air interaction methods. Wang and MacKenzie [22] found that the performance of freehand interaction with virtual objects significantly degraded when there was no physical surface to touch. Similarly, Vogel and Balakrishnan [23] found that the most significant challenge for conventional projector screens was the lack of a physical interaction plane for on-and-off gesture actions. The Midas touch problem is one of the most crucial technological challenges related to mid-air gesturing [24]. This happens when the gesture tracking technology recognizes unintentional user gestures as selections. These might be gestures connected with everyday actions, such as gesticulation during speech [14].

### 2.3. Mid-Air Interaction with Large Fogscreens

Fogscreens as a transparent physical interface can function as a touchscreen or create virtual- or mixed-reality environments when paired with suitable sensors. Conventional off-the-shelf 2D/3D tracking methods usually need modifications for a fogscreen because of the screen's immaterial nature or the visual disturbances caused by the fog [2]. Given this, and the fact that fogscreens are not common in research labs, the interaction properties of the fogscreen have not been studied systematically until now. Previously, Palovuori and Rakkolainen [3] implemented low-cost Microsoft-Kinect-based body tracking for interaction with mid-air projection screens for entertainment experiences, such as drawing applications and walk-through virtual reality. They reported that the Kinect tracker software did not detect users' fingers well enough due to the presence of the fog. In their next study, the tracking accuracy was improved; however, the technical specifications of the resulting system still did not allow physical interactions with the fog and did not reliably detect finger gestures inside of the fog. Their users performed hand gestures at a distance of 1–3 cm in front of the fog (with a Kinect device located behind the fog when facing the screen).

Other methods were applied to support interaction with various fogscreens. Plasencia et al. [25] merged an interactive tabletop screen with a vertical fogscreen in their MisTable system. Sand et al. [6] experimented with a small fogscreen with ultrasonic mid-air tactile feedback and found that participants preferred tactile feedback when selecting targets on a small fogscreen.

In summary, the available research on interaction with fogscreens is scattered and inconsistent, and there is a lack of user studies, particularly for physical manipulation within large fogscreens (when the hand or fingers are located inside the fog or touch the fog).

## 3. Interaction System Design

### 3.1. Screen and Projection

The interaction solution utilized a large vertical fogscreen coupled with the depth-sensing Microsoft Kinect Sensor V2 for Windows (https://developer.microsoft.com/en-us/windows/kinect, accessed on 21 June 2023) (see Figure 2). An early prototype of the fogscreen was used (see Jumisko-Pyykkö et al. [1], Rakkolainen et al. [5]). The fog was visible in an area roughly $1.55 \times 2.05$ m, but the lowest part was unusable for interaction due to strong fog fluctuations near the floor. The projected image ratio resulted in a $1.55 \times 1.05$ m area available for interaction.

The setup involved utilizing the Optoma ML750e mini LED projector, which has a contrast ratio of 15,000:1 and produces 700 ANSI lumens. The projector was positioned in front of the user, on the opposite side of the fog, as shown in Figure 2. This arrangement resulted in a brighter image compared to projecting from the user's side, as the user's body did not obstruct the projector's light.

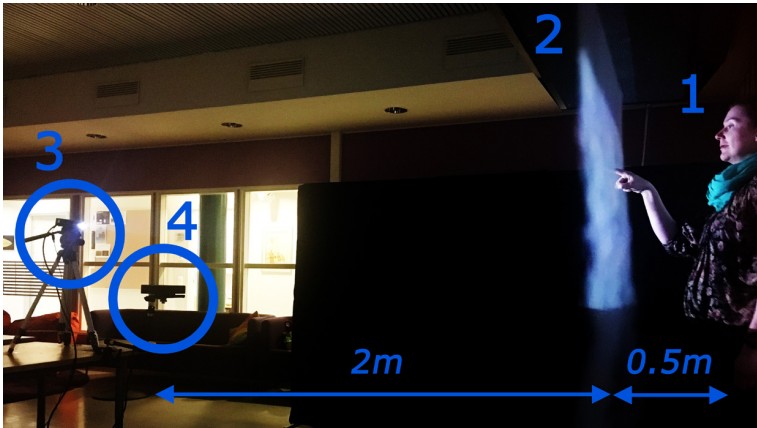

**Figure 2.** (1) User standing in front of the fogscreen; (2) fogscreen device; (3) projector; and (4) gesture sensor.

### 3.2. User Tracking

Placed behind the fogscreen, the Kinect device was tasked with monitoring the region directly in front of the screen. Utilizing the Kinect for Windows SDK 2.0 (https://www.microsoft.com/download/details.aspx?id=44561, accessed on 21 June 2023), the device tracked skeletal data of the user's hand while our software scrutinized hand gestures. The design required the user's index finger of the dominant hand to enter the fog. Thus, the system detected each entry/exit of the finger into/from the fog. A hand tip (usually detected as a fingertip of the index/middle finger of the right or left hand) controlled the mouse pointer. A four-point calibration matrix converted the depth vector obtained via the Kinect API (*CameraSpacePoint.Position*) into the screen point. The matrix contained Kinect vectors collected when pointing at the four corners of the fogscreen. The best possible calibration was used throughout the tests for all participants, as was a fixed geometry of the hardware setup (i.e., the Kinect and projector's positions). The calibration algorithm was as follows:

1. Initialize an empty buffer for 11 Kinect *CameraSpacePoint.Position* vectors;
2. Fill the buffer with vectors that are (a) "within" the fog (*CameraSpacePoint.Position.Z* < *Threshold*) and (b) far enough from the vector collected for the previous corner (except the first corner);
3. Compute medians of *X* and *Y* values of the vectors stored in the buffer—this will be the vector assigned to the corner being calibrated; and
4. Save the matrix of the four vectors into a file.

The screen point was then computed using the following algorithm:

1. Read the calibration matrix [TL, TR, BL, BR] from the file (the first calibration point TL corresponding to the origin of the screen serves as an offset);
2. Compute the average distance between screen edges in Kinect coordinates:
$size_X = ((TR_X - TL_X) + (BR_X - BL_X))/2$
$size_Y = ((BL_Y - TL_Y) + (BR_Y - TR_Y))/2$
3. Compute the X and Y scales as follows ($D = X|Y$):
$scale_D = ScreenResolution_D/size_D$
4. For each point received from Kinect, compute the screen point as:
$ScreenPoint_D = (KinectPoint_D - TL_D) * scale_D$

The algorithm was simple and robust, with a geometric setup used in the tests. The calibration quality was assessed manually. It was considered accurate if the mouse pointer was projected onto the index fingertip and could reach most locations around the fogscreen interaction area.

### 3.3. Haptic Device

To provide haptic feedback, we developed a lightweight wireless wearable vibrotactile actuation device. The design of the device was inspired by the haptic device created by Sand et al. [10]. Our design used an Arduino Nano microcontroller board in combination with an HC-05 serial Bluetooth radio and a Texas Instruments DRV2605L haptic motor controller, which incorporated an eccentric rotating mass (ERM) vibration motor. After extensive piloting, we chose a Bluetooth connection over a Wi-Fi connection due to erratic Wi-Fi signal delays caused by interference. To make the design wireless, the device was powered by two AA batteries regulated to 5V through a Pololu S7V8A stepper voltage regulator.

The device was attached to the participant's hand, with the motor controller around their wrist and the ERM motor to the index finger of the dominant hand on the palmar side, as seen in Figure 3. Although the placement of the actuator on the finger's tip or wrist might not affect performance, users favored having it on their finger in previous studies [1,18,20]. Moreover, research has shown that users responded faster when the hand used for target selection was stimulated with vibrotactile feedback [18].

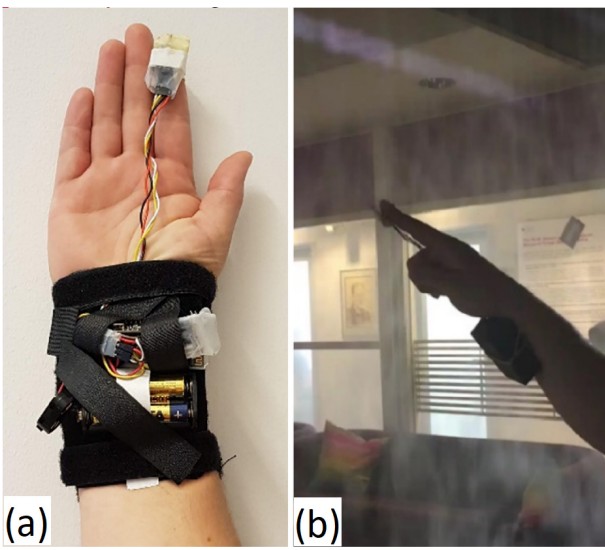

**Figure 3.** (**a**) Haptic device, (**b**) device in the experiment environment.

### 3.4. Interaction Gestures and Feedback

Visual feedback of the targets during interaction is shown in Figure 4. The targets for selection were blue circles on a black background (Figure 4a). The background effectively attenuated the bright spot of the projector (top-right corner of the pictures), which might otherwise create uneasiness for the participant's eyes. Both interaction gestures had two states: target acquisition and target selection.

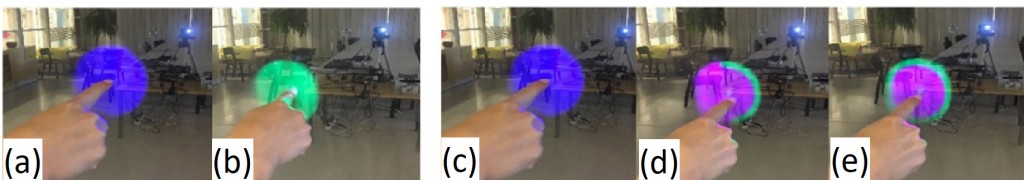

**Figure 4.** Visual feedback shown on the target: (**a**,**b**) tapping gesture; (**c**–**e**) dwell-based selection gesture.

The choice of tapping and dwell-based selection gestures was driven by earlier studies; while we are not aware of studies that performed experiments with a fogscreen, previous research with conventional large screens has shown that tapping and dwell-based gestures were found to be intuitive, accurate, and easy to use [13,14,17].

In fog tapping, the target acquisition state was triggered when the finger entered the target area, causing the target to turn green immediately (Figure 4b). As long as the finger remained inside the fog, it was possible to adjust its position. For instance, if the participant missed the target immediately after entering the fog, they could move their hand to relocate their finger on the target. To successfully execute the tapping gesture, the finger had to leave the fog within the target's current position boundaries. Upon achieving this, the target vanished, and a clicking sound confirmed the selection. A new target would then appear on the fog. However, if the finger exited the fog plane incorrectly, a unique error sound would play, indicating that the tapping gesture was not executed correctly.

During the target acquisition state of the dwell-based gesture, a green arc animation rotated around the target (Figure 4d) to indicate the progress of the dwell. The dwell time, set at 1000 ms, was determined through pilot testing and average values from previous studies [12,14,17,21]. Once the dwell time was completed, the target turned green and and disappeared, and a clicking sound was played while a new target appeared in a different location of the fog. If the target was left during the dwell, the dwell counter reset to its original duration, and the target turned blue. A small white cursor indicating the current position was available to help with target selection, and a new target was not shown until the current one was selected successfully for both gestures.

During fog tapping with haptic feedback, participants felt a sudden and intense vibrotactile sensation in their fingertip as the finger entered the fog, followed by a clicking sensation upon successfully selecting a target. No clicking feedback was given in the case of an incorrect target selection. In dwell-based selection, participants immediately received haptic feedback upon touching the target, followed by a continuous vibration that indicated the progress of the selection process. Once the selection was made, a strong clicking sensation was given, similar to that of fog tapping.

During the initial testing phase, the average duration of the fog tapping gesture was about 160 ms, while a comfortable duration of the dwell-based selection gesture was registered as 1000 ms. To ensure that the haptic feedback was easily recognizable, we utilized various characteristics of tactile-cue design, such as frequency, duration, rhythm, waveform, and location [26,27]. The vibrotactile stimulation patterns were carefully selected empirically within the frequency range of 150–300 Hz, which had been demonstrated to effectively stimulate Pacinian corpuscles in both glabrous and hairy skin regions, resulting in clear perception and recognition of haptic stimuli [28,29]. Finally, it was determined that a clicking sensation could be achieved through the use of short pulses lasting 60 ms at 200 Hz.

For the dwell-based gesture, a prolonged stimulus of 1000 ms was used. The stimulus was divided into four equal repetitions, inside which the frequency changed from 0 Hz to 200 Hz to 0 Hz over a duration of 250 ms. This was done to create rhythm in the longer stimulus.

## 4. Evaluation Using Fitts' Law

For an initial evaluation, the fogscreen was tested for basic gestural input for selecting targets. GoFitts (http://www.yorku.ca/mack/FittsLawSoftware/, accessed on 21 June 2023) was used for the experiment, which implements Fitts' law as per the methodology in the ISO 9241-9 standard for non-keyboard input devices [30]. The most common ISO 9241-9 evaluation procedure uses a 2D task with targets of width $W$ arranged in a circle. Selections proceed in a sequence moving across and around the circle (see Figure 5). Each movement covers an amplitude $A$—the diameter of the layout circle. The movement time ($MT$, in ms) is recorded for each trial and averaged over the sequence.

The difficulty of each trial is quantified using the index of difficulty ($ID$, in bits), and is calculated from $A$ and $W$ as:

$$ID = \log_2(A/W + 1) \qquad (1)$$

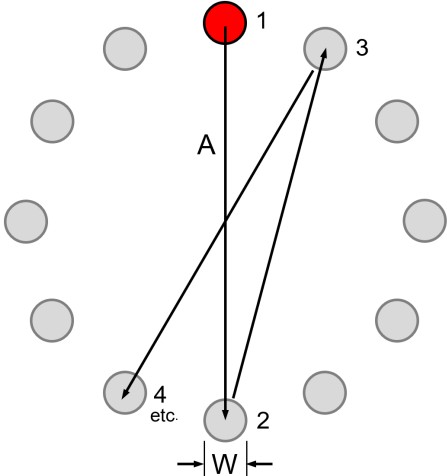

**Figure 5.** Two-dimensional target selection task in ISO 9241-9 [30].

The main performance measure in ISO 9241-9 is throughput (*TP*) in bits/second (bps), which is calculated over a sequence of trials as the *ID-MT* ratio:

$$TP = ID_e/MT \qquad (2)$$

The standard specifies calculating throughput using the effective index of difficulty ($ID_e$). The calculation includes an adjustment for accuracy to reflect the spatial variability in responses:

$$ID_e = \log_2(A_e/W_e + 1) \qquad (3)$$

with

$$W_e = 4.133 \times SD_x \qquad (4)$$

The term $SD_x$ is the standard deviation in the selection coordinates computed over a sequence of trials. For the 2D task, selections are projected onto the task axis, yielding a single normalized *x*-coordinate for each trial. For $x = 0$, the selection was on a line orthogonal to the task axis that intersects the center of the target. *x* is negative for selections on the near side of the target center and positive for selections on the far side. The factor 4.133 adjusts the target width for a nominal error rate of 4% under the assumption that the selection coordinates are normally distributed. The effective amplitude ($A_e$) is the actual distance traveled along the task axis. The use of $A_e$ instead of *A* is only necessary if there is an overall tendency for selections to overshoot or undershoot the target (see [31] for additional details).

Throughput is a potentially valuable measure of human performance because it embeds both the speed and accuracy of the user's responses. Therefore, comparisons between studies are possible, with the proviso that the studies use the same method in calculating throughputs. Figure 6 is an expanded formula for throughput, illustrating the presence of speed and accuracy in the calculation.

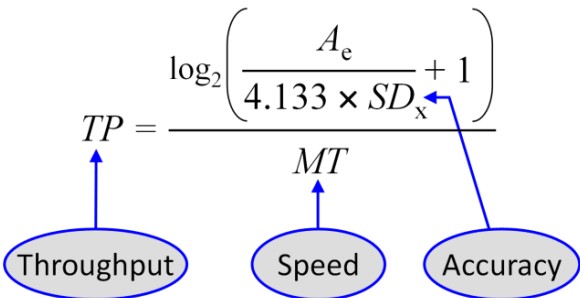

**Figure 6.** Expanded formula for throughput, featuring speed ($1/MT$) and accuracy ($SD_x$).

## 5. Methodology

### 5.1. Participants

Twenty healthy university students and staff members participated in the tests. Ages ranged from 20 to 66 years (mean = 35.3, $SD$ = 12.9). Some of the participants had prior experiences with mid-air gesturing (e.g., VR/AR goggles, Kinect-based gaming). Some had knowledge of immaterial screens (e.g., they had seen fogscreens in shopping malls or research demonstrations). None of the participants had first-hand experience of fogscreen interaction.

### 5.2. Apparatus

The experiment took place in a large open room with no direct sunlight. The lighting to improve the projected image was dimmed to a level where it was still comfortable to move about and somewhat comfortable to read and answer the questionnaire. The evaluation used the hardware setup as described in Section 3. The GoFitts software described in Section 4 provided the Fitts' law task and data collection. The tests involved two experimenters. One instructed the participants and the other acted as a software operator. During the test, both experimenters sat by the table and observed the participant from about a 3 m distance. The performance of the target acquisition task was monitored from a laptop computer with the following specifications: MSI GT72S Dominator Pro with Windows 10, 2.7 GHz Intel Core i7-6820HK, 16 GB RAM. The 1920 × 1080 pixel display was projected onto the fogscreen, with 10 cm corresponding roughly to 100 pixels.

### 5.3. Procedure

Participants arrived at the laboratory, read an information sheet, gave informed consent, and completed a background form. After that, the first gestural technique was explained, and they watched a video (Supplementary Materials) of the selection method in question. Next, participants were given instructions on how to use the fogscreen. They were told to keep their index finger perpendicular to the fog when selecting the targets. In addition, they were told to take a small step horizontally towards the more distant targets before selecting them, if needed. After the video and instructions, the gesture was demonstrated by the experimenter and followed by enough practice trials for the participant to feel comfortable with the interaction.

After the practice trials, the participants were told to make selections as fast and as accurately as possible, but at a comfortable pace, and that they could take extra breaks between the trial sequences, if needed. After a brief relaxation period, testing began. Figure 1 shows a participant interacting with the fogscreen.

The four conditions combining the two selection methods with the two feedback methods were presented in a counterbalanced order to offset learning effects. The haptic device was placed on the participant's hand only for selection methods with haptic feedback and it was removed when the selection methods were without haptic feedback. After each condition, participants were asked to evaluate the condition on a rating scale. The ratings were given with ten nine-point bipolar subjective scales modified from the NASA Task Load Index (TLX) [32]. The scales varied from a negative (−4) to positive (4) experience. Ratings

were provided for *general evaluation* (poor–good), *pleasantness* (unpleasant–pleasant), *quickness* (slow–quick), *accuracy* (inaccurate–accurate), *physical demand* (difficult–easy), *mental demand* (difficult–easy), *temporal demand* (high–low), *frustration* (high–low), *distractibility* (difficult–easy), and *usability/applicability* (unusable–usable). The scales were explained to the user if needed.

After the experiment, a final rating scale and a free-form questionnaire about using the gestural techniques with the fogscreen were provided. Participants were asked to rank the four conditions in order of preference from the most (1) to the least (4) preferred. Testing took about 1 hour for each participant. As an incentive, participants received a movie theater voucher.

*5.4. Design*

The primary independent variables were the selection method and feedback mode. Thus, the experiment was a two-way within-subject design with 2 selection methods (tapping, dwelling) $\times$ 2 feedback modes (audio-visual, audio-visual + haptic). Target amplitude and target width were varied by 100, 350, and 600 pixels, and 40, 70, and 100 pixels, respectively. Fitts' index of difficulty ranged from $ID = \log_2(100/100 + 1) = 1.0$ bits to $ID = \log_2(600/40 + 1) = 4.0$ bits. For each target $A$-$W$ condition, 15 targets were presented in a layout circle, as described earlier (see Figure 5).

The dependent variables were throughput (bps), movement time (ms), and error rate (%). We also recorded target re-entries, which is the number of times the finger pointer re-entered the target after the first entry. Regression models were built to test the interaction for conformance to Fitts' law. Each participant completed four blocks, consisting of two selection methods with two feedback modes. Within each block, participants performed 15 trials for each of the three target amplitudes and three target widths. With 20 participants, the total number of trials was $20 \times 2 \times 2 \times 3 \times 3 \times 15 = 10{,}800$.

*5.5. Data Pre-Processing, Outlier Removal, and Data Analysis*

A preliminary analysis revealed some anomalous behaviors that resulted in deviating responses—outliers. For the tapping condition, participants sometimes misselected the targets by inadvertently placing their hand in the fog during a movement. This caused an unintended selection at a coordinate far away from the intended target. Trials in which the selection coordinate deviation was more than $3\times$ the radius from the target center were deemed outliers and removed. In all, 0.9% trials met this criterion.

Using the dwell-based gesture, participants occasionally took a very long time before meeting the dwell interval of 1 second. Usually, this occurred with small targets or near the beginning of testing while participants were getting accustomed to the dwell selection. Trials where the movement time exceeded 10 s were deemed outliers and removed. In all, 48 trials met this criterion. In this way, the total number of outlier trials was 144, resulting in the removal of 1.3% of the original 10,800 trials. All analyses below are based on the data with outliers removed. A 2 (selection method) $\times$ 2 (feedback mode) within-subjects analysis of variance was performed on the data combined for all amplitudes and widths. Bonferroni-corrected *t*-tests were used for pairwise post hoc comparisons.

A Friedman test was used to compare the subjective ratings. In case of a statistically significant effect, the Wilcoxon signed-ranked test with Bonferroni correction was conducted for pairwise comparisons, resulting in a significance level threshold of $p < 0.008$ at an alpha level of 0.05 [33].

## 6. Results

*6.1. Throughput*

The grand mean for throughput was 1.81 bps. The highest throughput was 2.26 bps for tapping selection with audio-visual feedback. The lowest throughput was 1.40 bps for dwell-based selection with audio-visual feedback (see Figure 7a). A two-way ANOVA showed a statistically significant main effect of the feedback on throughput ($F_{1,19} = 109.0$,

$p < 0.0001$), while the main effect of the feedback was not statistically significant ($F_{1,19} = 0.92$, ns). The interaction of the main effects of the selection methods and feedback modes on throughput was also significant ($F_{1,19} = 4.91$, $p = 0.039$).

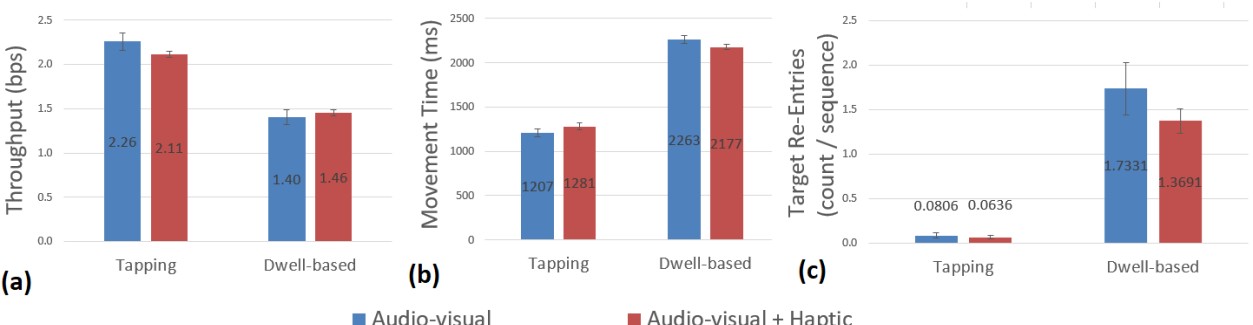

**Figure 7.** Plots of mean values: (**a**) throughput, (**b**) movement time, and (**c**) target re-entries by selection method and feedback mode. Error bars represent ±1 standard error of the means (SEMs).

Because of the significant interaction of the main effects of selection methods and feedback mode, one-way repeated measures ANOVAs were performed for both feedback modes and selection methods separately. ANOVA showed a significant effect of the selection method with the audio-visual feedback ($F_{1,19} = 113.8$, $p < 0.001$) and with audio-visual–haptic feedback ($F_{1,19} = 53.95$, $p < 0.001$). However, there were no significant differences between one selection method with different feedback modes. Pairwise post hoc comparisons showed that throughput was significantly higher for tapping with audio-visual feedback than for the dwell-based gesture with audio-visual feedback ($t' = 10.67$, $df = 19$, $p < 0.001$) and for tapping with audio-visual–haptic feedback than for the dwell-based gesture with audio-visual–haptic feedback ($t' = 7.35$, $df = 19$, $p < 0.001$).

### 6.2. Movement Time

The grand mean for movement time was 1732 ms. The highest movement time was 2263 ms for dwell-based selection with audio-visual feedback. The lowest movement time was 1207 ms for tapping selection with audio-visual feedback (see Figure 7b). Two-way ANOVA showed a statistically significant main effect of the selection method on movement time ($F_{1,19} = 622.2$, $p < 0.0001$), while the main effect of feedback mode on movement time was not statistically significant ($F_{1,19} = 0.03$, $p > 0.05$). The interaction of the main effects of the selection method and feedback mode on movement time was also significant ($F_{1,19} = 6.13$, $p = 0.023$).

Because of the significant interaction of the main effects of selection methods and feedback mode, one-way repeated measures ANOVAs were performed for both feedback modes and selection methods separately. ANOVA showed a significant effect of the selection method with the audio-visual feedback ($F_{1,19} = 439.0$, $p < 0.001$) and with audio-visual–haptic feedback ($F_{1,19} = 307.1$, $p < 0.001$). However, there were no significant differences within one selection method with different feedback modes. Pairwise post hoc comparisons showed that tapping with audio-visual feedback was significantly faster than dwell-based selection with audio-visual feedback ($t' = 20.95$, $df = 19$, $p < 0.001$) and tapping with audio-visual–haptic feedback was significantly faster than dwell-based selection with audio-visual–haptic feedback ($t' = 17.52$, $df = 19$, $p < 0.001$).

### 6.3. Target Re-Entries

The grand mean for target re-entries was 0.81 per sequence of 15 targets. The highest target re-entries were 1.73 for dwell-based selection with audio-visual feedback. The lowest target re-entries were 0.06 for tapping selection with audio-visual and haptic feedback (see Figure 7c). Two-way ANOVA showed a statistically significant main effect of the selection method on target re-entries ($F_{1,19} = 65.99$, $p < 0.0001$), while the main effect of the

feedback mode on target re-entries was not statistically significant ($F_{1,19} = 2.18$, $p > 0.05$). The interaction of the main effects of the selection method and the feedback mode on target re-entries was not significant ($F_{1,19} = 4.91$, $p = 0.039$).

Post hoc comparisons showed that there were significantly smaller numbers of re-entries for tapping with audio-visual feedback than for dwell-based selection with audio-visual feedback ($t' = 5.87$, $df = 19$, $p < 0.001$) or with audio-visual–haptic feedback ($t' = 10.93$, $df = 19$, $p < 0.001$). Furthermore, re-entries were significantly fewer for tapping with audio-visual–haptic feedback than for dwell-based selection with audio-visual feedback ($t' = -5.70$, $df = 19$, $p < 0.001$) or with audio-visual–haptic feedback ($t' = -9.83$, $df = 19$, $p < 0.001$).

*6.4. Fitts' Law Models*

Linear regression was performed to find models for each gesture. The results are shown in Figure 8, showing the intercept and slope as well as correlations of the model for each gesture and feedback modality. The correlations were generally high, indicating the overall conformance to Fitts' law (see Table 2).

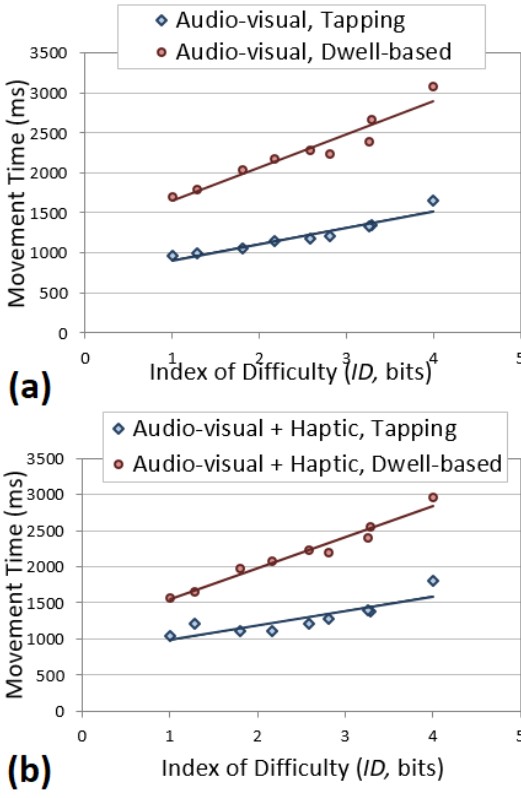

**Figure 8.** Linear regression between the movement time (*MT*) and index of difficulty (*ID*) for tapping and dwell-based gestures (**a**) without haptic feedback and (**b**) with haptic feedback.

**Table 2.** Fitts' law model for each gesture.

| Selection Mode | Haptic Feedback | *a* (Intercept) | *b* (Slope) | Correl (*r*) |
| --- | --- | --- | --- | --- |
| Tapping | no | 205.53 | 700.29 | 0.9142 |
| Dwell-based | no | 411.08 | 1248.8 | 0.9229 |
| Tapping | yes | 200.13 | 787.87 | 0.7434 |
| Dwell-based | yes | 429.71 | 1117.1 | 0.9664 |

### 6.5. Subjective Ratings

The means preference ratings are shown in Figure 9. The Friedman test showed a statistically significant effect for participants' preferences in interaction gesture, $\chi^2(3) = 7.980$, $p = 0.046$. Post hoc pairwise comparisons with Wilcoxon signed-rank tests showed that dwell-based selection with audio-visual–haptic feedback was significantly preferred over dwell-based selection with audio-visual feedback, $Z = -2.870$, $p = 0.004$. Other pairwise comparisons were not statistically significant. Figure 10 shows the results of the questionnaire. Table 3 presents the mean ranks of other subjective ratings. There was a statistically significant effect in participants' subjective rating for general evaluation, $\chi^2(3) = 7.971$, $p = 0.047$, and quickness, $\chi^2(3) = 18.57$, $p < 0.001$, of interaction gestures. Post hoc pairwise comparisons with Wilcoxon signed-rank tests showed that tapping with audio-visual feedback was preferred over dwell-based selection with audio-visual feedback, $Z = -2.686$, $p = 0.007$, and with audio-visual–haptic feedback, $Z = -3.224$, $p = 0.001$, and that tapping with audio-visual–haptic feedback was faster than dwell-based selection with audio-visual feedback, $Z = -2.734$, $p = 0.006$, and with audio-visual–haptic feedback, $Z = -3.119$, $p = 0.002$. There were no other significant differences in the subjective ratings. Table 4 lists some participants' comments on their experience with fogscreen. All the participants reported the intensity of the tactile feedback as sufficient; one said it even felt too intense.

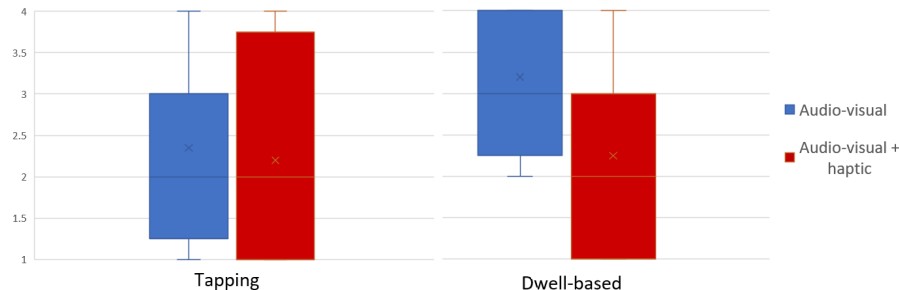

**Figure 9.** Participants' mean preference ratings for the selection methods. A lower score is better.

**Table 3.** Mean ranks of rating scales for all interaction gestures (scales from −4 to 4).

|  | Tapping | Dwell-Based | Tapping + Haptic | Dwell-Based + Haptic | Significance, *p* |
|---|---|---|---|---|---|
| General evaluation | 2.45 | 2.00 | 2.95 | 2.60 | <0.05 |
| Pleasantness | 2.40 | 2.25 | 2.68 | 2.68 | ns |
| Quickness | 2.93 | 2.05 | 3.18 | 1.85 | <0.001 |
| Accuracy | 2.45 | 2.15 | 2.68 | 2.73 | ns |
| Physical demand | 2.65 | 2.38 | 2.83 | 2.15 | ns |
| Mental demand | 2.65 | 2.58 | 2.45 | 2.33 | ns |
| Temporal demand | 2.80 | 2.50 | 1.98 | 2.73 | ns |
| Frustration | 2.78 | 2.38 | 2.20 | 2.65 | ns |
| Distractibility | 2.45 | 2.00 | 2.80 | 2.75 | ns |
| Usability/applicability | 2.58 | 2.35 | 2.75 | 2.33 | ns |

**Table 4.** Original comments from participants.

| | |
|---|---|
| **Interaction by tapping gesture** | *'It was not so fatiguing;' 'it was the easiest and most comfortable;' 'I was not getting tired;' 'it was more natural and fast;' 'I felt much like with iPad'* |
| **Interaction by dwell-based gesture** | *'It was too slow;' 'it was unpleasant;' 'it took too much time'* |
| **Haptic feedback** | *'Haptic feedback gave me some support;' 'haptic in tapping distracted;' 'good-to-have;' 'dwell-based gesture improved feedback;' 'more fun and exciting'* |
| **Interaction with the fogscreen** | *'Continued trials meant my finger got cold;' 'bottom targets were difficult to see and touch when crouching down;' 'the experience felt exciting'* |

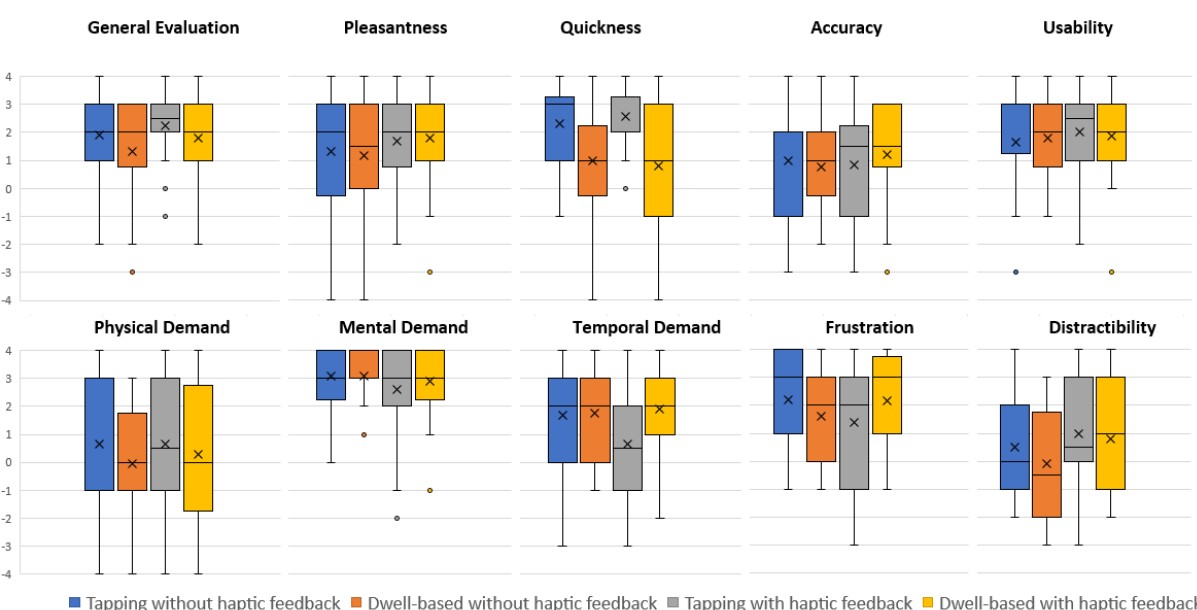

**Figure 10.** Subjective rating of participants' preference for all interaction gestures, with mean markers represented by × and outlier points.

## 7. Discussion

The comparisons of mid-air gestural selection methods, both with and without haptic feedback, using the 2D Fitts' law task provided interesting findings. It was confirmed that both gestures, tapping and dwell-based, performed reasonably well with a large fogscreen with throughput values ranging from 1.4 bps (dwelling) to 2.26 bps (tapping), as Figure 7a shows. This study represents one of the early attempts to systematically investigate the interactive properties of large fogscreens. We are unaware of other user studies of fogscreen interaction to which we can compare our empirical results directly. In the following, we compare our results to similar systems where mid-air gesturing was used for selection tasks.

Thus, our results for throughput values for mid-air tapping with fog are 32% higher than the 1.75 bps reported previously for pure mid-air tapping [15]. Furthermore, the results are comparable with the values of 2.10–2.25 bps for mid-air gesture selection with large conventional displays, where selection confirmation was executed by voice or button press [18,19] but lower than the values of 4–6 bps for large touchscreen interaction [34,35]. Regarding dwell-based selection with fog, its throughput is also comparable to the values of 1.7–1.9 bps previously reported [15,21]. These numbers allow us to assume that a tapping gesture performed in the fog is simpler for users to carry out than a tapping gesture performed solely in the air. The main difference is perhaps in the fact that our participants had a visual reference as to where the interaction should occur (e.g., the actually visible wall of fog) and they were able to estimate how far they needed to reach to make a tapping gesture.

In our study, tapping yielded an approximately 60% higher throughput, was faster, and produced fewer re-entries than dwell-based selection. A higher throughput may be explained by the fast task completion time of the tapping gesture, while the dwell-based selection gesture had a delay of 1000 ms. The speed superiority of the tapping gesture compared to the dwell-based gesture (500–800 ms) has been reported in previous studies [12,15]. A low number of re-entries for the tapping gesture was not expected because other researchers that compared tapping and dwell-based gestures in interaction with conventional screens have reported opposite findings [12,15]. It happened because in this study it was possible to readjust the finger position inside the fog while performing the tapping gesture. In contrast, the dwell-based gesture had a significantly higher number of re-entries compared to the tapping gesture. This could be attributed to hand jitter during the delay period, which may cause unintentional movement of the finger outside the target.

Subjective ratings were overall positive, with distractibility having the lowest value (see Figure 10). Mental demand was generally rated low, meaning that both the system and interaction were perceived as easy to use. We anticipated a more significant physical demand, but that too was on the positive side of the scale. The reason for the low physical demand was likely the fogscreen's visual feedback, which helped faster and easier orientation of the hand in the air, and this in turn required less effort.

Interestingly, participants ranked tapping with haptic feedback as the most preferred interaction method and dwell-based selection with haptic feedback as the second most preferred interaction method (see Figure 7). Overall, even though tapping was most preferred and gave the best throughput, subjectively, no method was rated as significantly better over the others. Most participants commented that tapping was faster, more pleasant, more accurate, and aligned with their previous experiences with touchscreens. At the same time, dwell-based gestures were considered slow and unpleasant. However, participants noted that dwell-based gestures were helpful in the fogscreen parts that were hard to touch (top of the screen) or with higher turbulences and less visibility (bottom of the screen). Additionally, some participants commented that continuous interaction with the fog using dwell-based gestures made their fingers cold. The overall preference for tapping is likely explained by the perceived delay when using dwell-based gestures, which required participants to hold their hand still for one second. As found in previous research [17,36], dwell-based gestures were tiring but practical if the recognition quality of the other gestures was poor.

The inclusion of haptic feedback did not have a significant impact on the measured performance of participants. We suggest that the effect of haptic feedback on the performance of mid-air interaction with a fogscreen depends on the selection method used; while previous research by Dube et al. [15] reported that haptic feedback improved performance across all interaction methods, our study revealed more nuanced outcomes. In our studies for the tapping gesture, we did not observe significant differences in performance metrics between conditions with and without haptic feedback. This implies that haptic feedback may not consistently enhance performance for this particular gesture in the context of mid-air selection with a fogscreen. It is possible that the visual feedback with the fogscreen and the audio feedback were sufficient for interaction and compensated for the absence of tactile feedback. However, further investigation is necessary to confirm this speculation. Pairwise comparisons between visual and auditory feedback are needed to see how haptic feedback fares and whether it can be substituted for feedback from other modalities. In the subjective ratings, participants somewhat preferred the inclusion of haptic feedback with the dwell-based gesture. This likely happened because fog and other common light-reflecting particles reduced the hand tracking accuracy. Moreover, many participants seemed to mistrust gestural interfaces initially, and haptic feedback could reassure the user that the system is indeed tracking the selections reliably. These notions, taken together with the tracking issues in the extreme distances and the intuitiveness of the dwell time-based selection gesture [13], underline that dwell time-based selection with haptic feedback can contribute to improved user experience, even if it does not contribute towards performance.

Finally, participants had an exciting experience and were having fun while interacting with the fogscreen. Participants commented that fog interaction by tapping gesture felt similar to interactions with touchscreens. However, there were some difficulties with small targets on the top and bottom parts of the screen. Thus, participants needed to make additional movements, such as crouching, stretching, or even standing on their toes. These body behaviors have also been noted previously [2,4]. It would be interesting to study the performance differences that result from using different body behaviors and movement strategies in target selection, in order to inform the design of bigger fogscreen user interfaces.

Our work here focuses on the evaluation of the interaction performance of two mid-air gestural selection methods, both with audio-visual-haptic and audio-visual feedback. It is important to note we did not investigate audio feedback independently, but we

incorporated auditory cues as an additional modality to enhance the user experience with fogscreens and provide multi-sensory feedback during mid-air gestural interaction. Moreover, previous studies on the effect of auditory feedback on mid-air gesture interaction showed that such feedback did not significantly affect task completion performance [37]. However, we acknowledge the importance of understanding the individual contributions of auditory feedback in future research.

Putting together the findings on throughput, movement time, target re-entries, and subjective ratings, it seems that tapping without haptic feedback was the best hand selection gesture for physical interaction with a fogscreen. Thus, these findings are consistent with the previous findings of Sand et al. [10]. Notably, our findings hold value, as prior research on mid-air gesture interaction has been critical of tapping selection gestures, has not explicitly demonstrated the excellence of these specific pushing forward gestures, and has recommended care in their use [13,21]. The tapping gesture could be used for rapid interactions, such as writing text or selecting menu items. In contrast, the slower dwell-based technique could be applied to operations that require more effort from the user, such as confirmation tasks or shutting down the system. On the other hand, the results showed that the performance of mid-air gestures with a fogscreen can be superior to that with conventional large screens. Furthermore, the inclusion of haptic feedback in the tapping gesture did not improve the performance. However, haptic feedback may be useful for dwell-based selection in situations where the user needs to interact with parts of the fogscreen with lower visibility. Future studies may need to explore alternative techniques for providing haptic feedback with a device attached to the user's hand. Based on the inherent properties of fogscreens, we posit that they can provide a genuinely contactless means of interaction that may prove beneficial for shared public use of fogscreens. This contactless experience could be particularly valuable in the context of the COVID-19 pandemic, where reducing physical contact can help mitigate the spread of the virus. However, when developing an application that involves interaction with a fogscreen, it is advisable to consider using the middle part of the screen to prevent users from having to make unnecessary movements. Overall, careful consideration of these factors can help to improve the user experience and effectiveness of fogscreen interaction.

## 8. Conclusions

We studied gestural interaction with large immaterial fogscreens using Fitts' law as an evaluation method. Fog tapping and dwell-based selection were compared with and without vibrotactile feedback. In summary, the results showed a statistically significant difference in Fitts'-law-based throughput values, favoring tapping over dwell-based gestures. Both quantitative data and subjective ratings and opinions show that tapping without haptic feedback is a prospective selection method for interaction with fogscreens, as it is natural and fast, and corresponds to users' previous experiences with touchscreens. Furthermore, it is not necessary to use haptic feedback for effective mid-air interaction with a large fogscreen, as a visual reference is sufficient.

The current results can be utilized in human–computer interaction and applied to the development of applications based on physical interaction with fogscreens in place of touchscreens.

**Supplementary Materials:** The following are available online at https://www.mdpi.com/article/10.3390/mti7070063/s1.

**Author Contributions:** Conceptualization, V.R., A.S., I.S.M., V.S. and Y.G.; methodology, V.R., A.S., I.S.M., O.Š., V.S. and Y.G.; software, V.R., A.S., I.S.M. and O.Š.; formal analysis, V.R., A.S., I.S.M. and V.S.; investigation and data curation, V.R., A.S., K.N. and O.Š.; writing—original draft preparation, V.R., A.S., I.S.M. and O.Š.; writing—review and editing, V.S., I.R., K.N., A.K. and Y.G.; supervision, V.S. and Y.G.; project administration, V.S. and Y.G.; funding acquisition, V.S., A.K. and Y.G. All authors have read and agreed to the published version of the manuscript.

**Funding:** The Faculty of Information Technology and Communication Sciences at Tampere University: funding for the doctoral studies of Vera Remizova; The Research Council of Finland: grant #326430; The Research Council of Finland: grant #351492.

**Institutional Review Board Statement:** Not applicable.

**Informed Consent Statement:** Informed consent was obtained from all subjects involved in the study.

**Data Availability Statement:** The underlying experimental data used to support the findings of this study are available from the corresponding author upon request.

**Acknowledgments:** We thank all the participants, all publications support, and staff, who wrote and provided helpful comments on previous versions of this document.

**Conflicts of Interest:** The authors declare no conflict of interest.

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
