# Peer review of "Mid-Air Gestural Interaction with a Large Fogscreen"

_mti, doi:10.3390/mti7070063_

Round 1

Reviewer 1 Report

The paper presents an user study for mid-air selection with a fogscreen. The study has a low novelty as it seems to mostly test a subset of the conditions tested in [6]. There is the use of auditory feedback, but not tested separately from the visual one.

This makes not possible to understand if the lack of effects of haptic feedback is due to the use of the auditory feedback, of the fog screen features, or of the different haptic feedback technology. This is a relevant issue of the experimental design but I don't know if it can be fixed with further experiments (it would add value to the paper)

The results of [6] reported here seem also not correct, as the throughput for tap in [6] is 2.27 not 1.75.

Also concerning the role of haptic feedback, the sentence "This contrasts with prior research by Dube et al. [6] who reported that haptic feedback improved performance across all interaction methods" is not correct as for tapping it seems that there are no significant differences in throughput with and without feedback and a I think authors could add some tests without auditory feedback if possible, if not they should at least point out the differences with respect to [6] an correct the wrong results and comments.

As one of the main results (tapping better than dwell-based gesture) was already found in other works and it expected, authors should point out what is the novelty in the proposed experiment and in the results.

The English is sufficiently good and the paper is well written

Author Response

Dear reviewer,

We would like to thank you for your careful and thorough reading, and for the questions and the constructive comments, which helped us to improve this paper. We also appreciate your critical remarks and will necessarily take them into account in future work. In response to your questions, we have provided detailed comments below each question, addressing the specific concerns and providing further clarification where needed. 

We would like to mention that the text changes in the manuscript have been marked in blue color.

Sincerely,

Authors

The paper presents an user study for mid-air selection with a fogscreen. The study has a low novelty as it seems to mostly test a subset of the conditions tested in [6].

The authors’ answer: We understand your concern regarding the novelty of our study. While it is true that our work builds upon previous research, specifically (Dube et al., 2022), we would like to highlight the unique contributions and different experimental conditions that our study offers. Our work focuses specifically on mid-air selection techniques for a fogscreen, which presents its own set of challenges and opportunities. We believe that investigating the impact of mid-air interaction with this particular display technology is valuable, as it expands the knowledge base for designing interactive systems using fogscreens.

Furthermore, our study investigates a subset of the conditions tested in [6] to allow for a controlled comparison and a more in-depth analysis of the mid-air selection techniques specifically applicable to fogscreens. We believe this focused approach contributes to the understanding of mid-air interaction within this unique context.

There is the use of auditory feedback, but not tested separately from the visual one.

This makes not possible to understand if the lack of effects of haptic feedback is due to the use of the auditory feedback, of the fog screen features, or of the different haptic feedback technology. This is a relevant issue of the experimental design but I don't know if it can be fixed with further experiments (it would add value to the paper)

The authors’ answer: We appreciate your suggestion to evaluate auditory feedback separately from visual feedback. While our study did not explicitly test auditory feedback in isolation, we incorporated auditory cues as an additional modality to enhance the user experience and provide multi-sensory feedback during mid-air selection tasks. The primary goal of our study was to investigate the performance of mid-air hand gestures for interaction with a fogscreen, considering the combination of both visual and auditory feedback.

However, we acknowledge the importance of understanding the individual contributions of auditory feedback in future research. Based on your suggestion, we plan to conduct further investigations to evaluate the impact of auditory feedback independently, in order to gain a more comprehensive understanding of its potential benefits and drawbacks.

The following sentences were added to the text: (Section 7, 8th paragraph, line 501) “Our work here focuses on the evaluation of the interaction performance of two mid-air gestural selection methods, both with audio-visual-haptic and audio-visual feedback. It is important to note we did not investigate audio feedback independently, but we incorporated auditory cues as an additional modality to enhance the user experience with fogscreens and provide multi-sensory feedback during mid-air gestural interaction. Moreover, previous studies on the effect of auditory feedback on mid-air gesture interaction showed that such feedback did not significantly affect task completion performance [19]. However, we acknowledge the importance of understanding the individual contributions of auditory feedback in future research.”

We also added the following reference:

19. Köpsel, A., Majaranta, P., Isokoski, P., and Huckauf, A. (2016) Effects of auditory, haptic and visual feedback on performing gestures by gaze or by hand. In Behaviour & Information Technology, 35:12, 1044-1062.

The results of [6] reported here seem also not correct, as the throughput for tap in [6] is 2.27 not 1.75.

The authors’ answer: We appreciate your observation regarding a discrepancy in the reported results of (Dube et al., 2022). The misunderstanding arose due to the usage of similar names for different gestures in our work. In our manuscript, we employed unified names for mid-air gestural techniques to facilitate a clear understanding and comparison across different studies. That was mentioned in table notes on page 4: “The table uses unified names for mid-air gestural techniques. Tapping refers to a pushing forward gesture moving a certain distance, which is also known as ‘push’ or ‘tap’ in the literature.”

To ensure consistency in our comparisons, we specifically compared our throughput values with the corresponding gesture in (Dube et al., 2022), which they referred to as the "push" gesture. The reported throughput value of 1.75 in our paper corresponds to the "push" gesture in (Dube et al., 2022). We apologize for any confusion caused by the mismatch in terminologies and appreciate your attention to this detail.

Also concerning the role of haptic feedback, the sentence "This contrasts with prior research by Dube et al. [6] who reported that haptic feedback improved performance across all interaction methods" is not correct as for tapping it seems that there are no significant differences in throughput with and without feedback and a I think authors could add some tests without auditory feedback if possible, if not they should at least point out the differences with respect to [6] an correct the wrong results and comments.

The authors’ answer:  We appreciate your detailed notion and the opportunity to address the concerns raised. We apologize for any inaccuracies in our statement and would like to provide the necessary clarifications. Regarding the sentence stating that “This contrasts with prior research by Dube et al. [6]", we acknowledge that our interpretation was not entirely accurate. We apologize for the confusion caused by this misrepresentation of the results from (Dube et al., 2022).

The following sentences were added to the text: (Section 7, 6th paragraph, line 480) “We suggest that the effect of haptic feedback on the performance of mid-air interaction with a fogscreen depends on the selection method used. While previous research by Dube et al. [7] reported that haptic feedback improved performance across all interaction methods, our study revealed more nuanced outcomes. In our studies for the tapping gesture, we did not observe significant differences in performance metrics between conditions with and without haptic feedback. This implies that haptic feedback may not consistently enhance performance for this particular gesture in the context of mid-air selection with a fogscreen.”

As one of the main results (tapping better than dwell-based gesture) was already found in other works and it expected, authors should point out what is the novelty in the proposed experiment and in the results

The authors’ answer:  Thank you for your valuable feedback on our paper regarding the comparison between tapping and dwell-based gesture in mid-air selection. While it is true that previous works have reported the superiority of tapping over dwell-based gesture in mid-air selection, we would like to emphasize the specific novelty and contributions of our study.

Firstly, the novelty lies in the application of these mid-air gestural selection methods within the unique context of interacting with a fogscreen. The fogscreen technology offers distinct opportunities for interaction compared to other display systems. By evaluating the tapping and dwell-based gestures specifically in this fogscreen context, we provide insights into their performance and effectiveness within this specific environment.

Secondly, in our work, tapping refers to a pushing forward gesture for a certain distance, which is also known as "push" or "tap" in the literature. It is important to note that previous studies have not explicitly demonstrated the superiority of this specific pushing forward gesture, which we refer to as tapping, in mid-air gestural interaction.

The following sentences were added to the text: (Section 7, 6th paragraph, line 506) “…has not explicitly demonstrated the excellence of these specific pushing forward gestures…”

Reviewer 2 Report

Please to kindly read the attached file.

Author Response

Dear reviewer,

Thank you for your positive feedback on the manuscript and your suggestion for improvement, which help us to improve this paper. In response to your questions, we have provided detailed comments below each question, addressing the specific concerns and providing further clarification where needed. 

We would like to mention that the text changes in the manuscript have been marked in blue color.

Sincerely,

Authors

I suggest including A, and W in the Figure 5 to a better understanding of that parameters. It costs low effort and the graph gains in clarity.

The authors’ answer: Thank you for your positive feedback on the manuscript and your suggestion for improvement. In response to your suggestion, we have revised Figure 5 to incorporate parameters A and W as you recommended. By doing so, we believe that the graph now provides a more comprehensive representation of the relevant data, allowing readers to better understand the relationships between the variables.

I think is not clear enough this sentence (lines 324-325): “With 20 participants, the total number of trials was 20 × 2 × 2 × 3 × 3 × 15 = 10,800”. A more in detail explanation of the meaning of these factors and conditions of the trial is mandatory.

The authors’ answer: I apologize for the lack of clarity in the previous explanation. We revised and added a more detailed explanation of the factors and conditions of the trial.

The following sentences were added to the text: (Section 5.4, 2nd paragraph, line 324) “Each participant completed four blocks, consisting of two selection methods with two feedback modes. Within each block, participants performed 15 trials for each of the three target amplitudes and three target widths.”

In line 343, a specific technique is used but no reference is cited: “…the Wilcoxon signed-ranked test with Bonferroni correction…” I think it would be recommended to give more details or propose references to justify this assumption.

The authors’ answer: Thank you for pointing out the lack of reference for the specific technique used for subjective ratings analysis. To address this concern, we will revise the manuscript to include a suitable reference that justifies the use of this statistical analysis.

We added the following reference:

6. Demšar J. (2006) Statistical Comparisons of Classifiers over Multiple Data Sets. In Journal of Machine Learning Research 7 1–30.

Format defects (abnormal: font size/style, line breaks, acronyms, …) Line 21: there appears cites 15, 20 and 21 before the 1-14. It is very usual to number cites in appearing order of the manuscript avoiding jumps.

The authors’ answer: Thank you for your comment regarding the numbering of the references in the manuscript. We appreciate your feedback and understand the concern about the order in which the references appear. We would like to clarify that the numbering of the references was done in accordance with the journal's recommendations, which suggested using alphabetic order for the references rather than the order of appearance in the manuscript. We followed this guidance to ensure consistency with the journal's citation style and to align with common practices in the field.

Reviewer 3 Report

The authors present a topic of interest, namely Mid-Air Gestural Interaction, which represents a stage in the realization of the Brain Computer Interface.

The authors have a rich experience in approaching this topic, and the article is very well structured and argued. I have no ambiguities from the point of view of the methodology and the solution of the problem addressed.

I have only one suggestion for the authors - to introduce in chapter 7 Discussions a comparison between their method and other similar methods from the specialized literature.

Author Response

Dear reviewer,

Thank you for your positive feedback on our article. We appreciate your recognition of our experience and the overall structure and argumentation of the paper. We also appreciate your suggestion to include a comparison with other similar methods from the specialized literature. We agree that providing a comparative analysis of our method with other existing techniques will further enhance the depth and significance of our work. Our study represents one of the early attempts to systematically investigate the interactive properties of large fogscreens. We are unaware of other user studies of fogscreen interaction to which we could compare our empirical results directly. However, we do discuss and compare our results to similar systems in which mid-air gesturing was used for selection tasks. We hope this is a sufficient comparison. 

We would like to mention that the text changes in the manuscript have been marked in blue color.

Sincerely,

Authors

Round 2

Reviewer 1 Report

Authors tried to fix some issues, others cannot be without doing novel experiments.

Reviewer 2 Report

I would like to thank authors for considering my suggestions in the revised manuscript.